# Neutral CB1 Receptor Antagonists as Pharmacotherapies for Substance Use Disorders: Rationale, Evidence, and Challenge

**DOI:** 10.3390/cells11203262

**Published:** 2022-10-17

**Authors:** Omar Soler-Cedeno, Zheng-Xiong Xi

**Affiliations:** Addiction Biology Unit, Medicinal Chemistry Section, Molecular Targets and Medications Discovery Branch, Intramural Research Program, National Institute on Drug Abuse, Baltimore, MD 21224, USA

**Keywords:** cannabinoid, CB1 receptor, Δ^9^-tetrahydrocannabinol, rimonabant, PIMSR, AM4113, neutral antagonist, inverse agonist, substance use disorders

## Abstract

Cannabinoid receptor 1 (CB1R) has been one of the major targets in medication development for treating substance use disorders (SUDs). Early studies indicated that rimonabant, a selective CB1R antagonist with an inverse agonist profile, was highly promising as a therapeutic for SUDs. However, its adverse side effects, such as depression and suicidality, led to its withdrawal from clinical trials worldwide in 2008. Consequently, much research interest shifted to developing neutral CB1R antagonists based on the recognition that rimonabant’s side effects may be related to its inverse agonist profile. In this article, we first review rimonabant’s research background as a potential pharmacotherapy for SUDs. Then, we discuss the possible mechanisms underlying its therapeutic anti-addictive effects *versus* its adverse effects. Lastly, we discuss the rationale for developing neutral CB1R antagonists as potential treatments for SUDs, the supporting evidence in recent research, and the challenges of this strategy. We conclude that developing neutral CB1R antagonists without inverse agonist profile may represent attractive strategies for the treatment of SUDs.

## 1. Introduction

Substance use disorder (SUD), defined as the uncontrollable and persistent use of drugs (including alcohol) despite substantial harm and adverse consequences, is still a severe social and health problem worldwide. SUD-related costs, including those in crimes, loss of productivity and healthcare, exceed $740 billion per year in the Unites States [1]. In recent years, opioid overdose and SUD-related diseases have increased dramatically with the fatal incidents up to ~50,000 in 2017 in the USA [2]. Although the United States Food and Drug Administration (FDA) approved several medications such as methadone, buprenorphine, and varenicline for the treatment of opioid or nicotine use disorders [3,4,5], the rate of relapse remains extremely high. Moreover, there is no FDA-approved medication for the treatment of psychostimulant use disorders [6]. Over the past decades, the cannabinoid receptor 1 (CB1R) has been given much attention as a promising target in medication development for treating SUDs [7,8,9]. The reason for such attention is because of convincing evidence indicating that rimonabant, a selective CB1R antagonist with an inverse agonist profile, is highly effective in reducing drug taking and drug-seeking behavior in experimental animals [7,8,10]. However, the severe adverse effects of rimonabant, such as nausea, emesis, depression, and suicidal tendencies observed in humans have led to its withdrawal from clinical trials worldwide [11]. Consequently, the US FDA decided not to approve CB1R ligands until better safety and efficacy data become available. In this mini-review article, we first review the rationale and supporting evidence for developing rimonabant and its analogs as promising pharmacotherapies for SUDs, and then discuss the possible mechanisms underlying rimonabant’s therapeutic benefits and unwanted side-effects. Lastly, we discuss the recent research progress and the challenges in developing neutral CB1 receptor antagonists as new pharmacotherapies for SUDs.

## 2. Mesocorticolimbic Dopamine System

To better understand how cannabinoid CB1R antagonists produce anti-addictive effects, it is necessary to briefly review the current working hypothesis underlying drug reward and addiction. Addiction includes three stages—binge/intoxication, a stage at which an individual consumes an intoxicating substance and experiences its rewarding effects; withdrawal/negative affect, a stage at which an individual experiences a negative emotional state in the absence of the substance; and preoccupation/anticipation, a stage at which subject seeks substances again after a period of abstinence [12]. Although our understanding of the neural mechanisms underlying each stage of addiction is still not fully understood, a well-accepted view is that the rewarding effects of drugs of abuse are mediated mainly by activation of the mesocorticolimbic dopamine (DA) system. This system originates in DA neurons in the ventral tegmental area (VTA) and substantia nigra pars compacta (SNc) of the midbrain and projects to the prefrontal cortex (PFC), nucleus accumbens (NAc), and the dorsal striatum (SD) [6,13]. Different drugs of abuse activate this pathway by distinct receptor and cellular mechanisms [14,15,16] (Figure 1). For example, the psychostimulant cocaine activates this system mainly by blocking the DA transporter (DAT), while nicotine activates VTA DA neurons by stimulating nicotinic receptors located on DA neurons or glutamate neurons that project to DA neurons in the VTA and NAc [16,17,18]. Alcohol’s reinforcement has been associated with processes involving multiple molecular targets, including mu opioid receptors and NMDA receptors [12,19,20,21]. On the other hand, opioids activate midbrain DA neurons mainly by stimulation of opioid receptors located on GABAergic neurons in the rostromedial tegmentum (RMTg) and substantia nigra pars reticulata (SNr) that project to the VTA and SNc, respectively, causing increases in DA neurons firing and striatal DA release via GABA-mediated disinhibition [5,17,22]. Therefore, both the RMTg-VTA-NAc and SNr-SNc-DS DA pathways play a central role in drug reward and addiction [14,22], making the DA system a crucial target in medication development for the treatment of SUDs.

## 3. Endocannabinoid System

To better understand how CB1R antagonists produce therapeutic effects against drug abuse and addiction and how CB1R inverse agonists produce unwanted side-effects, let us briefly review the endocannabinoid (eCB) system and recent research on how cannabinoids modulate the mesocorticolimbic DA system, a critical action site for drugs of abuse.

The endocannabinoid (eCB) system consists of cannabinoid receptors (CB1Rs, CB2Rs, and others), endocannabinoids [anandamide (AEA) and 2-arachidonoylglycerol (2-AG)], enzymes for endocannabinoid synthesis [*N*-arachidonoyl phosphatidylethanolamine-phospholipase D (NAPE-PLD), diacylglycerol-lipase (DAG-lipase)] and degradation [fatty acid amide hydrolase (FAAH) and monoacylglycerol lipase (MGL)], and their putative transport systems [23,24]. AEA was the first endocannabinoid discovered by Raphael Mechoulam and his colleagues in 1992 [25]. AEA is an endogenous CB1R agonist (Ki = 87.7 nM for rCB1; Ki = 239.2 nM for hCB1) and a weak CB2R agonist (Ki = 267.8 nM for rCB2; Ki = 439.5 nM for hCB2) [26]. The effects of AEA are mediated mainly by activation of CB1Rs and CB2Rs in the brain and periphery. However, AEA levels in the brain are very low in healthy subjects, and it has a very short half-life (~2 min) due to its fast degradation by fatty acid amide hydroxylase (FAAH) [27]. Therefore, the functional significance of AEA in the brain is largely unclear.

2-AG was the second eCB discovered in the brain [28]. It is an endogenous agonist of the CB1Rs (Ki = 1180 nM for rCB1; Ki = 3423 nM for hCB1) and CB2Rs (Ki = 1900 nM for rCB2; Ki = 1193 nM for hCB2) [26]. Unlike AEA, 2-AG is present at relatively high levels in the central nervous system (CNS). Therefore, it is thought to be a major eCB modulating brain function.

There are at least two types of cannabinoid receptors (CB1Rs and CB2Rs) identified in the brain [23]. The phytocannabinoids (Δ^9^-THC), synthetic cannabinoids (WIN55,212-2, CP55,940, HU-210), and the endocannabinoids (AEA, 2-AG) all have high binding affinities at both the CB1Rs and CB2Rs [23]. Cannabinoids may also bind to other putative cannabinoid receptors, such as G protein-coupled receptor 55 (GPR55), transient receptor potential vanilloid 1 (TRPV1) channel, and peroxisome proliferator-activated nuclear receptors (PPARs) [23]. Accumulative evidence indicates that cannabinoid action is mediated mainly by activation of CB1Rs and CB2Rs [23].

## 4. Cannabinoid Reward *versus* Aversion

Cannabis is the most commonly used substance worldwide as many people find it pleasurable [29]. However, the findings regarding the rewarding properties of cannabinoids in both humans and experimental animals are conflicting [23]. Indeed, cannabis use has often been associated with its psychoactive, rewarding effects [30,31]. The psychoactive effects of cannabis, combined with the ongoing cannabis legalization in the United States, may well explain why cannabis use is rising in the USA. For instance, from 2002 to 2019, the percentage of adults who reported using cannabis in the past year increased from 7.0 to 15.2% [32].

However, cannabis enjoyment is not universal, and some individuals report dysphoria, anxiety, and depression after cannabis use [33,34]. The increase in cannabis use also raises concerns about possible adverse effects of cannabis use, such as developing the amotivational syndrome [35,36], which is defined as “a reduction in the motivation to initiate or persist in goal-directed behavior” [37]. A series of human functional magnetic resonance imaging (fMRI) studies support these cannabis amotivational effects by evidence that Δ^9^-THC produces a significant reduction in reward-related brain activity or neural response to reward in healthy adults [38,39,40]. In congruent with these findings, other reports showed that Δ^9^-THC reduced the likelihood or motivation of reward-related learning and decision-making [41], dampened neural responses to music [42], and reduced striatal DA response to reward [43].

Similar paradoxical effects of cannabinoids have been discovered in non-human primates, as squirrel monkeys self-administer Δ^9^-THC or endocannabinoids [44,45], while other primate species (rhesus, baboon, cynomolgus) fail to demonstrate this behavior [46,47,48]. In rodents, Δ^9^-THC alone is not self-administered [49,50], although the mixture of Δ^9^-THC and cannabidiol was recently reported to be self-administered by rats [51,52]. In conditioned place preference (CPP) test, Δ^9^-THC typically produces conditioned place aversion [53,54], although place preferences have also been reported [55,56]. In electrical intracranial self-stimulation (ICSS) experiments, Δ^9^-THC was initially reported to facilitate electrical ICSS in rats [56,57,58], while other studies found suppression of ICSS in rats and mice [59,60,61,62,63]. In optogenetic ICSS (oICSS) maintained by optical stimulation of midbrain DA neurons or glutamate neurons, cannabinoids always produce a reduction in brain-stimulation reward (BSR) in mice, suggesting a reward-attenuating or aversive effect [64,65].

Similarly, the findings of cannabinoid action on DA transmission are also conflicting. There are reports indicating that activation of the CB1Rs increases DA neuronal firing in the VTA [66,67] and DA release in the NAc in rats [68,69,70,71]. However, *in vitro* voltammetry experiments in striatal brain slices demonstrate that the cannabinoids WIN55,212-2 or CP55,940 fail to alter [72,73] or produce a reduction in electrical stimulation-induced DA release in the dorsal striatum in guinea pigs, rats and mice [74,75]. *In vivo* microdialysis experiments in freely moving animals indicate that Δ^9^-THC produces a dose-dependent reduction in NAc DA in mice [76]. The neural mechanisms underlying such opposite affective and neurochemical effects of cannabinoids are not fully understood.

### 4.1. GABAergic CB1R Hypothesis of Cannabis Reward

Given that midbrain DA neurons receive both inhibitory GABAergic and excitatory glutamatergic inputs, we proposed that differential CB1R expression on GABAergic neurons *versus* glutamatergic neurons may underlie cannabinoid reward *versus* aversion, respectively [7,23,64] (Figure 2). The GABAergic CB1R hypothesis is supported by electrophysiological findings in brain slices where stimulation of CB1Rs on VTA GABAergic neurons causes an increase in VTA DA neuron firing via GABA-mediated disinhibition [77,78,79,80]. However, so far, there is a lack of behavioral evidence *in vivo* supporting this GABA-CB1R hypothesis possibly due to the absence of reliable behavioral models of cannabinoid reward in rodents.

### 4.2. Glutamatergic CB1 Hypothesis of Cannabinoid Aversion

Clearly, the above GABAergic disinhibition hypothesis cannot explain how cannabinoids produce the aversive effects observed in rodents. To address this question, we have recently used advanced RNAscope *in situ* hybridization (ISH) assays to examine the cellular distributions of CB1Rs. We found that CB1Rs are expressed not only in VTA GABAergic neurons but also in VTA glutamatergic neurons [23,64,65]. Strikingly, optogenetic activation of VTA glutamatergic neurons produced potent rewarding effects, as assessed by CPP and optical ICSS (oICSS) [64,81]. Systemic administration of multiple cannabinoids (such as Δ^9^-THC, WIN55,212-2, ACEA, AM-2201) dose-dependently inhibited glutamate-mediated oICSS only in VgluT2-Cre control mice, but not in glutamate-CB1-knockout mice in which CB1Rs are selectively deleted from subcortical VgluT2-expressing glutamate neurons [64]. These findings suggest that activation of CB1Rs on glutamate neurons produces reward-attenuation or aversive effects by decreasing glutamatergic inputs onto VTA DA neurons (Figure 2).

The above findings also suggest that activation of brain CB1Rs is not always rewarding but it could be aversive, depending upon the cellular distribution of CB1R expression in the brain. Therefore, we propose that the hedonic effects of cannabis might depend on the balance of two opposing actions of cannabinoids on both GABAergic neurons and glutamatergic neurons (Figure 2). If more CB1Rs are expressed in VTA or VTA-projecting GABAergic neurons, cannabis will be rewarding as GABAergic disinhibition of VTA DA neurons is dominant. In contrast, if more CB1Rs are expressed in VTA or VTA-projecting glutamatergic neurons, cannabis will be aversive as CB1R mediated reduction in glutamatergic inputs onto VTA DA neurons is dominant. Congruently, if CB1R levels are equivalent on both types of neurons, cannabis should have no net effect on the brain reward function (Figure 2). This hypothesis appears to well explain why Δ^9^-THC or cannabinoids are rewarding in some human subjects and non-human primates (squirrel monkeys) in which more CB1Rs might be expressed in VTA-projecting GABA neurons but are ineffective or aversive in other human subjects and species (such as rats, mice) in which more CB1Rs might be expressed in VTA or VTA-projecting glutamate neurons (Figure 2).

### 4.3. Dopaminergic CB2 Hypothesis of Cannabinoid Aversion

In addition to the above glutamate-CB1R hypothesis, recent research indicates that CB2Rs are also expressed in VTA DA neurons and contribute to the aversive effects of cannabinoids [3,82]. Activation of CB2Rs by JWH133 inhibits VTA DA neuron activity and decreases NAc DA release in wildtype mice, but not in CB2-KO mice [83,84,85,86]. Activation of CB2Rs or overexpression of brain CB2Rs also inhibits cocaine self-administration, cocaine-induced CPP and hyperactivity in mice [85,87,88,89]. In rats, Δ^9^-THC and WIN55, 212-2 produce biphasic effects on electrical brain-stimulation reward (BSR)—reward-enhancing at lower doses and reward-attenuating (or aversive) at higher doses [63]. CB1R antagonism (by AM251) reduced the low dose-enhanced BSR, while CB2R antagonism (AM630) decreased the high dose-attenuated BSR. Congruently, selective CB1R and CB2R agonists produced significant BSR enhancement and inhibition, respectively [63]. Together, these findings suggest that DA-CB2R mechanisms, at least in part, underlie cannabinoid-induced aversion (Figure 2) [3,23]. Thus, the subjective effects of cannabinoids would depend on the balance of multiple cell type-specific receptor mechanisms. This cell type-specific cannabinoid receptor mechanism appears to well explain why cannabis or cannabinoids could be rewarding, aversive, or ineffective since the cellular distributions of CB1Rs and CB2Rs may be different in different subjects or species.

## 5. Rimonabant: The First CB1R Antagonist Approved for the Treatment of Obesity

*Rimonabant discovery*: Extensive studies in the past decades indicate that endocannabinoids are overactive in obese humans [90,91,92] and obese animals in both genetic and diet-induced obesity [93,94], which inspired research to develop CB1R antagonists for the treatment of obesity. The initial effort failed in developing a selective CB1R antagonist from modifications of the structure of Δ^9^-THC until 1994, when Sanofi Pharmaceutical Inc. in France developed rimonabant (also called SR141716A, trade names Acomplia, Zimulti) as the first CB1R antagonist used for the treatment of obesity [95].

In June 2006, the European Commission approved the sale of rimonabant in the then-25-member European Union as a prescription drug for use in conjunction with diet and exercise for overweight patients with a body mass index (BMI) > 30 kg/m^2^ or patients with a BMI > 27 kg/m^2^ with associated risk factors, such as type 2 diabetes or dyslipidemia [96]. It was the world’s first approved anti-obesity drug in this class. By 2008, rimonabant was available in 56 countries. Data from clinical trials showed that rimonabant had severe adverse effects—causing depressive disorders or mood alterations in up to 10% of subjects and suicidal ideation in around 1%. Other common adverse effects included nausea, vomiting, and upper respiratory tract infections in more than 10% of patients [97,98,99]. Post-marketing surveillance data found that the risk of psychiatric disorders in people taking rimonabant was doubled [100]. Rimonabant was submitted to the U.S. FDA for approval in 2005. In 2007, the FDA concluded that Sanofi-Aventis failed to demonstrate the safety of rimonabant and voted against recommending the anti-obesity treatment for approval [101]. Two weeks after FDA’s decision, the company withdrew the application. In October 2008, the European Medicines Agency recommended suspension of clinical use of rimonabant after concluding that its risks outweighed its benefits, and its approval was withdrawn by the European Commission in January 2009 [100]). As a consequence, clinical trials with rimonabant and almost all other CB1R antagonists with similar inverse agonist profiles such as the longer acting second generation surinabant (SR147778, Sanofi-Aventis), ibipinabant (SLV319, Solvay Pharmaceutical), taranabant (MK-0364, Merck), and otenabant (CP-945,598, Pfizer) [102] were terminated worldwide in 2008 [97,99].

Rimonabant’s failure is a sad and disappointing story in medication development for the treatment of obesity as clinical trial data indicate that rimonabant had significant beneficial effects in controlling body weight and obesity [103] and significant improvement in glycemic control and lipid profile in type 2 diabetic patients [104,105]. Unfortunately, rimonabant’s psychiatric side-effects, such as anxiety, depression, and suicidality [98], led to its withdrawal from clinical trials and the drug market. However, we should point out that some confounders could be implicated, such as the inclusion of obese patients with a previous history of depression, a fact that could increase the possibility of detecting depressive symptomatology such as suicidal ideation or suicide in some patients treated with rimonabant [106,107]. In addition, when we consider rimonabant’s beneficial and adverse effects, it is also important to ask—what are the neural mechanisms underlying its therapeutic anti-obesity effects *versus* its psychiatric adverse effects?

*Pharmacology of rimonabant*: Structurally, rimonabant is an aminoalkylindole [108]. This compound shows high affinity for the CB1R (*K_i_* = 2 nM) but low affinity for the CB2R (K_i_ > 1000 nM). In vitro, rimonabant antagonizes the inhibitory effects of cannabinoid receptor agonists on both mouse vas deferens contractions and DA-stimulated adenylyl cyclase activities in rat brain membranes. After oral administration, rimonabant inhibited [^3^H]-CP55,940 binding to cerebral membranes with a median effective dose (ED_50_) value of 3.5 mg/kg [108]. Systemic administration (i.v.) of rimonabant inhibited the classical pharmacological effects of Δ^9^-THC such as hypoactivity, hypothermia, and antinociception in mice [109]. These data suggest that rimonabant is a selective CB1R antagonist both in vitro and in vivo.

Unexpectedly, it was also reported that rimonabant alone, at high dose (>3 mg/kg), produced increased locomotor activity in mice [109], an effect that is opposite to that produced by the cannabinoid receptor agonist Δ^9^-THC [95]. This finding was supported by observation that intrathecal injection of rimonabant also evoked a significant thermal hyperalgesia in mice, an effect also opposite to cannabinoid agonist-induced analgesia [110]. Congruently, [^35^S]GTP*γ*S binding assays in the cell membranes isolated from human CB1R-transfected Chinese hamster ovary (CHO) cells indicate that the cannabinoid agonist WIN 55,212-2 stimulated [^35^S]GTP*γ*S binding by ~80% above basal levels while rimonabant produced a >20% decrease in basal [^35^S]GTP*γ*S binding, suggesting that WIN 55,212-2 is a full CB1R agonist while rimonabant is an inverse agonist at CB1Rs [111] (Figure 3). These data indicate that rimonabant has dual actions—as a CB1R antagonist that blocks the activity of cannabinoid agonists on CB1Rs and as an inverse CB1R agonist by itself, producing an effect opposite to CB1R agonists in the absence of cannabinoids (Figure 3). This unique pharmacological action of rimonabant raised a fundamental question—whether CB1R antagonism or inverse agonism underlies the therapeutic anti-obesity *versus* the adverse effects of rimonabant. Clearly, understanding the receptor mechanisms underlying the therapeutic effects *versus* unwanted effects of rimonabant is critically important for successfully developing new generations of CB1R antagonists without significant adverse effects for the treatment of obesity and SUDs.

## 6. CB1R Antagonists Are Promising for the Treatments of SUDs

***Rationale***: In a series of clinical trials known as the Studies with Rimonabant and Tobacco Use (STRATUS), it was found that rimonabant significantly increased abstinence rates and reduced smoking cessation-related weight gain [9,112,113,114,115], suggesting that the endocannabinoid system may also be involved in nicotine use disorder and CB1R antagonists or inverse agonists may be useful for the treatment of SUDs including nicotine use disorder.

***Supporting evidence***: The findings with either CB1R agonists or antagonists support this hypothesis of eCB involvement in SUDs. For example, Δ^9^-THC increases heroin self-administration in rats [116]. WIN55,212-2 increases motivation to nicotine self-administration and facilitates cue-induced reinstatement of nicotine seeking in rats [117]. WIN55,212-2 or CP55,940 facilitate alcohol self-administration, CPP, and binge-like behavior in rodents [118,119]. Accordingly, it was proposed that CB1R antagonists should also be effective in the treatment of SUDs [8,120,121].

Indeed, compelling preclinical evidence supports this hypothesis. For example, rimonabant reduced intravenous heroin self-administration under fixed-ratio and progressive-ratio reinforcement [122,123], nicotine cue-induced reinstatement of nicotine-seeking behavior in rats [124,125], and nicotine-enhanced DA release in the NAc [124]. Furthermore, rimonabant also prevents the development of morphine-induced CPP [126], and dose-dependently attenuates heroin- or heroin-associated cue-induced reinstatement of drug-seeking behavior [123]. In addition, rimonabant blocks acquisition of cocaine-induced CPP [127] and attenuates reinstatement of drug seeking caused by cocaine or cocaine-associated cues [128]). Congruently, rimonabant also blocks reinstatement of methamphetamine-seeking behavior [129] and reduces alcohol intake in rodents [130]. These exciting findings with rimonabant encouraged many pharmaceutical industries to develop other brain-penetrant CB1R antagonists such as the longer acting second generation surinabant (SR147778, Sanofi-Aventis), taranabant (MK-0364, Merck), otenabant (CP-945,598, Pfizer), and ibipinabant (SLV319, Solvay Pharmaceutical), for the treatment of obesity, smoking and drugs of abuse [102].

***Receptor mechanisms:*** Unfortunately, all the above-mentioned ligands are not only CB1R antagonists but also inverse agonists. Thus, dissecting the role of CB1R antagonism *versus* inverse agonism in their therapeutic *versus* side-effects is critical for developing safer CB1R ligands for the treatment of SUDs.

***CB1R antagonism may underlie the therapeutic effects of rimonabant:*** Given that rimonabant is a well-characterized CB1R antagonist, it is reasonable to hypothesize that the CB1R antagonism may underlie its therapeutic anti-obesity and anti-addictive effects. This is supported by several lines of evidence. First, cannabinoids and drugs of abuse may act in a common neural substrate—the mesocorticolimbic DA system via distinct cellular and receptor mechanisms (Figure 2). Assuming that CB1Rs are tonically activated by eCBs (2-AG, AEA), CB1R antagonism on GABA neurons or GABAergic terminals would produce a reduction in eCB-enhanced DA transmission (Figure 2), which may functionally counteract the DA-enhancing and the pro-addictive effects produced by drugs of abuse. Second, growing evidence indicates that drugs of abuse (such as cocaine, heroin, or nicotine) may increase eCB release in the VTA and/or NAc [131,132,133,134,135,136,137,138] (Figure 4). After releasing from post-synaptic neurons, such as VTA DA neurons (Wang et al., 2015), eCBs retrogradely diffuse back to activate presynaptic CB1Rs, producing a reduction in neurotransmitter (GABA, glutamate) release, which subsequently produces reward-enhancing and addictive effects (Figure 4). Accordingly, CB1R antagonism at presynaptic terminals would block the eCB-mediated effects, producing anti-addictive effects (Figure 4). Third, the neutral CB1R antagonists without CB1R inverse agonist profile (such as AM4113 and PIMSR) produce the similar anti-addictive effects as rimonabant in drug self-administration and reinstatement, but without rimonabant-like depressive effects [7,13,139]. Lastly, in theory, rimonabant’s anhedonic or aversive effects may also functionally counteract the rewarding effects of drug abuse; however, direct supporting evidence is missing due to the absence of selective CB1R inverse agonists.

***CB1R inverse agonism may underlie the adverse effects of rimonabant:*** As discussed above, CB1R agonism by Δ^9^-THC on GABA neurons or GABAergic terminals may produce an increase in NAc DA release and reward-enhancing effects via a disinhibition mechanism (Figure 2 and Figure 5A). Accordingly, CB1R inverse activation by rimonabant on the same receptor would produce opposite effects—enhanced GABA and reduced NAc DA release, which may translate into depressive-like subjective effects of rimonabant (Figure 5B).

Three animal models are often used to evaluate the rewarding *versus* aversive effects [140,141]. They are intracranial self-stimulation (ICSS) (also called brain-stimulation reward, BSR), NAc DA response to test drugs, and conditioned place preference/aversion (CPP/CPA). The findings with rimonabant from these models are mixed (Table 1). In the ICSS model, we [139] and others [142] previously reported that high-dose rimonabant inhibits electrical ICSS, while three other reports [143,144,145] indicate that rimonabant has no effect on ICSS. With respect to DA response to rimonabant, we have previously reported that, in mice, systemic administration of rimonabant produces a reduction in extracellular DA in the NAc [146]. However, in Long-Evan rats, we found that systemic rimonabant failed to alter extracellular NAc DA, while intra-NAc local perfusion of rimonabant at 100 µM (but not at 0.1, 1.0, or 10.0 µM) unexpectedly increased extracellular NAc DA [76]. Another report indicates that—in Wistar rats—rimonabant produced enhanced NAc DA response [147]. In the CPP/CPA model, most publications reported that rimonabant, at low doses, produced neither CPP nor CPA (Table 1). Thus, although some evidence supports that rimonabant could be aversive by itself, which could be related to its inverse agonist profile, conclusive supporting evidence is still lacking. The negative findings may be related to rimonabant doses tested in the above studies and/or the distinct cellular distributions of CB1R expression in different species (rats vs. mice) or subjects under different experimental conditions. As stated below, the important findings with PIMSR, a neutral CB1R antagonists without inverse agonist profile, in the same animal models provide valuable evidence supporting an assumption that the CB1R inverse agonism at least in part underlie the adverse psychiatric effects of rimonabant, as discussed above.

## 7. Neutral CB1R Antagonists as New Promising Therapies for SUDs

Although withdrawn from the market, rimonabant still remains a valuable tool in cannabinoid research and in developing newer generation of CB1R ligands with different profiles, such as neutral CB1R antagonists without inverse agonist profile, peripherally restricted CB1R ligands, and allosteric CB1R modulators [102]. Notably, the pyrazole skeleton of rimonabant has been widely used as the starting point with side chain modifications in developing newer generations of CB1R neutral antagonists, leading to a series of pyrazole-based tricyclic ligands (such as PIMSR, VCHSR, AM4113, AM6527, NESS06SN, etc.) that display neutral CB1R antagonist profiles [102]. In addition, many other non-pyrazole-based ligands with neutral CB1R antagonist profile (such as thioamide, THCV, O-2654, O-2050, amauromine, and 018-gluc) have also been reported [102]. Here, we briefly review a few novel CB1R neutral CB1R antagonists that have been tested in experimental animal models of drug abuse.

**PIMSR:** PIMSR is a pyrazole derivative, which was designed computationally to stabilize both the active and inactive states of CB1R to afford neutral antagonism [152]. *In vitro* radioligand binding assays indicate that PIMSR has as high affinity (Ki  =  17–57 nM) for human CB1Rs expressed in cultured HEK cells as rimonabant does (Ki  =  1.8–18 nM) [152]. Computational molecular modeling (CB1R docking) and Ca^++^ channel assays indicate that PIMSR blocked WIN55,212-2-induced inhibition of Ca^++^ influx [152]. Electrophysiological assays indicate that co-administration of PIMSR reversed the inhibitory effects of Δ^9^-THC or synthetic cannabinoids (AM2201, AM018) on excitatory glutamate transmission in the hippocampus [153]. However, unlike rimonabant that increases Ca^++^ influx, PIMSR itself has no effect on the Ca^++^ influx [152], suggesting that PIMSR is a neutral CB1R antagonist without inverse agonist profile [154].

Systemic administration of PIMSR (10 mg/kg/day for 28 days) significantly reduces body weight and food intake in high-fat diet-induced obese mice [154]. We have recently evaluated the therapeutic potential of PIMSR against cocaine use disorder in experimental animals (Table 2). We found that systemic administration of PIMSR dose-dependently inhibited cocaine self-administration under fixed-ratio (FR1, FR5) reinforcement, shifted the cocaine self-administration dose-response curve downward, decreased incentive motivation to seek cocaine under progressive-ratio reinforcement, and reduced cue-induced reinstatement of cocaine seeking [13]. In addition, PIMSR dose-dependently attenuated cocaine-enhanced ICSS maintained by electrical stimulation of the medial forebrain bundle in rats. Importantly, PIMSR itself failed to alter electrical ICSS, which is often interpreted as a lack of reward-attenuation or depression-like effects (Table 2). This is further supported by our finding that PIMSR alone produced neither rewarding nor aversive effects in the CPP/CPA test. We also examined the effects of PIMSR on food-taking behavior in mice. We found that PIMSR dose-dependently inhibited oral sucrose self-administration and reduced sucrose intake. This finding suggests that PIMSR also retains rimonabant’s therapeutic anti-obesity effects.

**AM4113:** AM4113 is another novel CB1R neutral antagonist developed by Alexandros Makriyannis in 2007 [155]. Structurally, AM4113 is also a pyrazole-3-carboxamide analog of rimonabant [156]. In competitive [^3^H]-CP55,940 binding assays, AM4113 has a Ki value of 0.80 ± 0.44 nM and exhibits a 100-fold selectivity for CB1R over CB2R [157]. Unlike rimonabant, AM4113, at up to 10 µM concentration, has no effect on forskolin-stimulated cAMP accumulation in CB1R-transfected HEK-293 cells [157], suggesting a lack of an inverse agonist profile.

AM4113 also displayed positive therapeutic effects for SUDs in numerous behavioral tests (Table 2). Similar to rimonabant, AM4113 also inhibits nicotine self-administration, motivation for nicotine seeking, and nicotine priming-, nicotine-associated cue-, or stress-induced reinstatement of nicotine-seeking behavior in rats [158]. In consistency with these findings, we have recently reported that AM4113 dose-dependently inhibited heroin self-administration but was less effective at reducing cocaine or methamphetamine self-administration in rats under fixed-ratio reinforcement schedules [139]. In a similar way to AM251, pretreatment with AM4113 attenuated the aversive, affective effects of naloxone-precipitated morphine withdrawal in rats [159], suggesting an effect mediated by CB1R antagonism. In squirrel monkeys, both AM4113 and rimonabant attenuated nicotine- and Δ^9^-THC-seeking behaviors, as well as cue-induced reinstatement of cocaine seeking [160]. In addition, AM4113 reduced alcohol consumption and preference [161], and attenuated the discriminative effects of CB1R agonists [162]. Like rimonabant, AM4113 also reduced food intake and weight gain in rats, but not in CB1-KO mice [156,157,159] and precipitated cannabinoid withdrawal signs [163]. Together, these findings indicate that AM4113 retains rimonabant’s critical therapeutic effects for the treatment of SUDs and obesity (Table 2).

Importantly, AM4113 itself is devoid of many rimonabant-like untoward effects (Table 2). Unlike AM251, AM4113 did not potentiate vomiting in the ferret nor promote nausea [156,157]. Additionally, it did not produce malaise or anxiety-like effects [164]. AM4113 attenuated the depressive effects of WIN55,212–2 as assessed by behavioral profiles in open-field studies [164]. In the ICSS paradigm, rimonabant decreased electrical ICSS, while AM4114 did not [139], suggesting that AM4113 did not produce aversive or depression-like effects. All these data suggest that AM4113 has an improved safety profile over rimonabant. However, pharmacokinetic studies indicated that AM4113 had poor oral bioavailability [161]. In addition, AM4113 was reported to produce anxiety-like effects similar to that by rimonabant in an open field assay [164]. Thus, although AM4113 is still useful in the proof of concept study, the lack of oral bioavailability and the possible anxiety-like side-effects may limit its development as a clinical candidate [165].

**AM6527:** Unlike AM4113, AM6527 is an orally effective CB1R neutral antagonist [165]. AM6527 has high binding affinity to rat CB1R with ~100-fold selectivity for CB1R (Ki = 4.88 nM) over CB2Rs (Ki = 463) [165]. In experimental animals both AM6257 and AM4113 inhibited food-reinforced behavior under a FR5 reinforcement schedule with similar ED_50_ values (0.58 mg/kg vs. 0.78 mg/kg) after intraperitoneal administration. Notably, oral administration of AM6527 was also effective in attenuation of food intake with EC50 value of 1.49 mg/kg, while AM4113 was not effective at an oral dose up to 32 mg/kg [165]. Systemic administration of AM4113 or AM6527 prevented naloxone-precipitated morphine withdrawal in the CPA paradigm [166] (Table 2). In drug discriminative test, squirrel monkeys treated daily with the long-acting CB1R agonist AM2389 can effectively discriminate rimonabant from saline [167,168]. These discriminative-stimulus effects were both dose- and time-dependent and were stable for up to 48 days [167]. Importantly, antagonist substitution tests with the CB1R neutral antagonists AM4113 or AM6527 produced similar rimonabant-like discriminative effects, while agonist substitution tests with Δ^9^-THC (or nabilone, AM4054, JWH018, AM3506) reduced the discriminative effects of rimonabant [167], suggesting that both AM4113 and AM6527 retain rimonabant’s CB1R antagonist, but not inverse agonist profile. So far, the effects of AM6527 on drug self-administration and reinstatement of drug-seeking behavior have not been examined.

**Tetrahydrocannabivarin (THCV):** Δ^9^-Tetrahydrocannabivarin (Δ^9^-THCV), naturally found in *Cannabis*, is a homologue of Δ^9^-THC with a propyl side chain instead of a pentyl group. [^35^S]GTPγS binding studies indicated that Δ^9^-THCV acted as a CB1R neutral antagonist at low doses [169,170]. However, at higher doses, it may behave as a CB1 agonist, CB2R agonist or antagonist depending on the assays [170,171,172].

In experimental animals, Δ^9^-THCV produced hypophagic effects in both fasted and non-fasted mice [173]. Δ^8^-THCV is a synthetic, more stable, and easier-to-synthesize analogue of Δ^9^-THCV with a similar pharmacological receptor binding profile [171]. Systemic administration (i.p.) of Δ^8^-THCV significantly attenuated intravenous nicotine self-administration and cue-induced and nicotine-induced reinstatement of nicotine-seeking behavior in rats [174] (Table 2). Δ^8^-THCV also significantly attenuated nicotine-induced CPP and nicotine withdrawal symptoms in mice (Table 2), suggesting that Δ^8^-THCV or Δ^9^-THCV may have therapeutic potential for the treatment of SUDs [174].

**Table 2 cells-11-03262-t002:** Effects of CB1R neutral antagonists on drug-taking and drug-seeking behavior in experimental animals.

Compound	Doses	Species	Results	References
PIMSR	10, 30 mg/kg	Rat	↓ Cocaine self-administration (FR2, FR5, PR)	[13]
PIMSR	3, 10, 30 mg/kg, i.p.	Rat	↓ Cocaine-cue-induced reinstatement	[13]
PIMSR	3, 10, 30 mg/kg, i.p.	Rat	↓ Cocaine-enhanced electrical brain-stimulation reward	[13]
PIMSR	10, 30 mg/kg	Mouse	Not produce CPP or CPA by itself	[13]
PIMSR	3, 10, 30 mg/kg, i.p.	Rat	Not alter electrical brain-stimulation reward by itself	[13]
AM4113	3, 10 mg/kg, i.p.	Rat	Not alter cocaine self-administration	[139]
AM4113	0.3–3 mg/kg, i.m.	Monkey	Not alter cocaine self-administration	[160]
AM4113	0.3, 1, 3, 10 mg/kg, i.p.	Rat	↓ PR cocaine self-administration	[158]
AM4113	0.3–3 mg/kg, i.m.	Monkey	↓ Cue-induced cocaine seeking	[160]
AM4113	0.3–3 mg/kg, i.m.	Monkey	↓ Cocaine-primed drug seeking	[160]
AM4113	3, 10 mg/kg, i.p.	Rat	↓ Methamphetamine self-administration	[139]
AM4113	0.3, 1, 3, 10 mg/kg, i.p.	Rat	↓ Nicotine self-administration	[158]
AM4113	0.3–3 mg/kg, i.m.	Monkey	↓ Nicotine self-administration	[160]
AM4113	0.3, 1, 3, 10 mg/kg, i.p.;	Rat	↓ Cue-induced nicotine seeking	[158]
AM4113	0.3–3 mg/kg, i.m.	Monkey	↓ Cue-induced nicotine seeking	[160]
AM4113	0.3–3 mg/kg, i.m.	Monkey	↓ Drug priming-induced nicotine seeking	[160]
AM4113	0.3–3 mg/kg, i.m.	Monkey	↓ Δ^9^-THC self-administration	[160]
AM4113	0.3–3 mg/kg, i.m.	Monkey	↓ Cue-induced Δ^9^-THC seeking	[160]
AM4113	0.3–3 mg/kg, i.m.	Monkey	↓ Drug priming-induced Δ^9^-THC seeking	[160]
AM4113	3, 10 mg/kg, i.p.	Rat	↓ Heroin self-administration	[139]
AM4113	1, 2.5 mg/kg, i.p.	Rat	↓ Naloxone-precipitated CPA	[166]
AM4113	3, 10 mg/kg, i.p.	Rat	No effect on electrical brain-stimulation reward by itself	[139]
AM6527	1, 2.5 mg/kg, i.p.	Rat	↓ Naloxone-precipitated CPA	[166]
Δ^8^-THCV	10, 20 mg/kg, i.p.	Rat	↓ Nicotine self-administration	[174]
Δ^8^-THCV	0.03~3 mg/kg, i.p.	Mouse	↓ Nicotine-induced CPP	[174]
Δ^8^-THCV	10, 20 mg/kg, i.p.	Rat	↓ Nicotine- or cue-induced nicotine seeking	[174]
Δ^8^-THCV	0.3 mg/kg, i.p.	Mouse	↓ Nicotine withdrawal-induced somatic signs	[174]

In human clinical trials, Δ^9^-THCV treatment increased neural responding to rewarding and aversive stimuli [175]. Another human MRI study indicated that Δ^9^-THCV decreased resting state functional connectivity in the default mode network and increased connectivity in the cognitive control and dorsal visual stream networks and in the brain regions where functional connectivity is altered in obesity [176]. These findings suggest that Δ^9^-THCV deserve further research as a potential therapy for obesity and for SUDs.

## 8. Summary

The complexities of the neurobiology of SUDs create enormous challenges in efforts to develop effective pharmacological treatments. The initial finding of the overactive eCB status in human subjects with obesity led to the discovery of rimonabant, the first available CB1R antagonist/inverse agonist, for the treatment of obesity in humans. Subsequently, an unexpected finding that rimonabant, when used for the treatment of obesity in humans, also precipitated smoking cessation, inspired huge research effort to explore rimonabant and other CB1R antagonists for the treatment of obesity and SUDs. Although the results from both clinical and preclinical studies are promising, the adverse effects of rimonabant tampered initial enthusiasm and led almost all CB1R ligands withdrawn from human use and clinical trials. This unsuccessful story with rimonabant, although extremely disappointing, also encouraged research to investigate the neural mechanisms underlying the therapeutic *versus* adverse effects of rimonabant and inspired us to develop and test newer generations of neutral CB1R antagonist for the treatment of SUDs. Extensive research in this area has led to the discovery of numerous neutral antagonists in the past decade. However, only a few of them have been tested in experimental animals. The results demonstrate that the neutral antagonists such as PIMSR and AM4113 retain the therapeutic anti-addictive effects of rimonabant but without significant rimonabant-like adverse effects, which provides the first proof-of-concept evidence supporting that a purer (neutral) CB1R antagonist with little inverse agonism may have a more favorable pharmacological profile in the treatment of obesity and SUDs.

However, there are many challenges in this research area. The studies reviewed above may represent only an early stage in this strategy since it is highly challenging to really “dissect” or “separate” the pharmacological effects produced by CB1R antagonism *versus* inverse agonism, as both actions may functionally counteract the action produced by drugs of abuse. As inverse agonism, CB1R antagonism, in theory, may also produce unwanted side-effects. Paradoxically, rimonabant itself is not always aversive in the currently used ICSS and CPP/CPA behavioral models. So far, there is only limited evidence indicating that the neutral antagonists (PIMSR, AM4113) are lack of unwanted psychiatric effects. In addition, although preclinical findings with PIMSR and AM4113 have provided encouraging results, the relatively poor blood–brain penetration ability of PIMSR and poor oral bioavailability of AM4113 may lower their translational potential for the use in humans for the treatment of SUDs. Clearly, more research effort is required to either improve the pharmacokinetic profiles of the available CB1R antagonists (such as PIMSR and AM4113) or develop newer generations of neutral antagonists for the treatment of obesity and SUDs.

## Figures and Tables

**Figure 1 cells-11-03262-f001:**
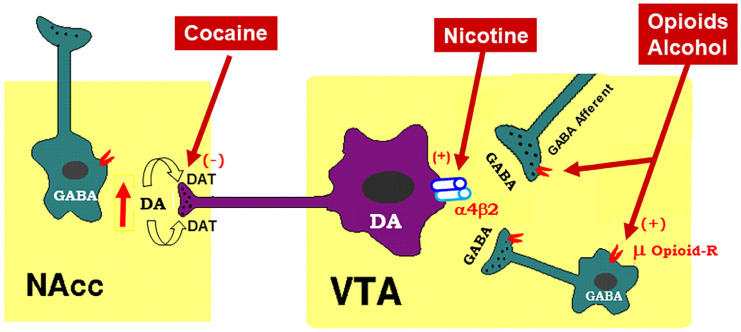
Schematic diagram of the mesolimbic dopamine (DA) hypothesis, illustrating how drugs of abuse activate this system. The mesolimbic DA system originates in the midbrain ventral tegmental area (VTA) and projects predominantly to the forebrain nucleus accumbens (NAc) and the prefrontal cortex (not shown). The psychostimulant cocaine elevates extracellular NAc DA by blocking DA transporters (DAT) on DA axon terminals, while opioids (such as heroin) and alcohol bind to and activate mu opioid receptor (μ Opioid–R) located mainly on GABAergic afferents (less on VTA GABAergic interneurons) and inhibit GABA release. A reduction in GABA release leads to DA neuron disinhibition (activation). Nicotine has been thought to activate DA neurons mainly by activation of α4β2 nicotinic receptors located on DA neurons. (+), (-): indicate activation of opiate receptors or blockade of DAT.

**Figure 2 cells-11-03262-f002:**
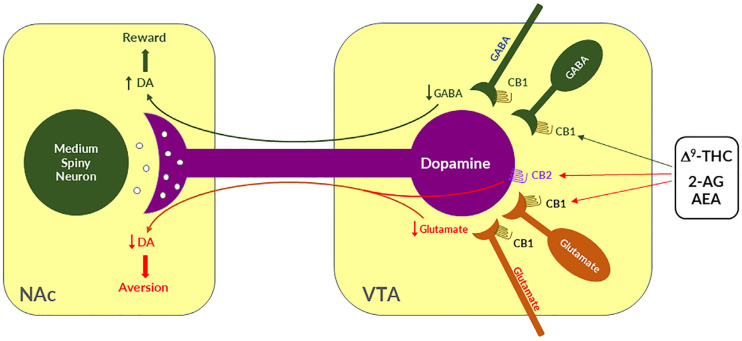
A schematic diagram showing the neural mechanisms underlying cannabis reward *versus* aversion. Cannabinoid CB1Rs are expressed not only in VTA GABAergic interneurons or GABAergic afferents but also in VTA glutamatergic neurons or their afferents within the VTA, whereas CB2Rs are also expressed in VTA DA neurons. Both CB1Rs and CB2Rs are inhibitory G-protein (Gi)-coupled receptors, producing an inhibitory effect on neuronal firing or terminal neurotransmitter release after activation. Cannabinoids such as Δ^9^-THC, 2-AG, and AEA may produce rewarding effects by binding to CB1Rs on VTA GABAergic interneurons and/or their afferents as a reduction in GABA release causes an increase DA neuronal firing and enhanced DA release in the NAc. Conversely, cannabinoids may also produce aversive effects by activating CB1Rs on glutamatergic neurons and/or terminals in the VTA that decreases excitatory glutamate input on VTA DA neurons. In addition, activation of CB2Rs on VTA DA neurons also produce an inhibitory effect on DA neuron firing and DA release in the NAc. Thus, the subjective effects of cannabinoids may depend on the balance of both oppose actions. This hypothesis may well explain why cannabinoids are rewarding in some subjects or species, while ineffective or even aversive in others. ↑, ↓—indicate an increase or a decrease in neurotransmitter release.

**Figure 3 cells-11-03262-f003:**
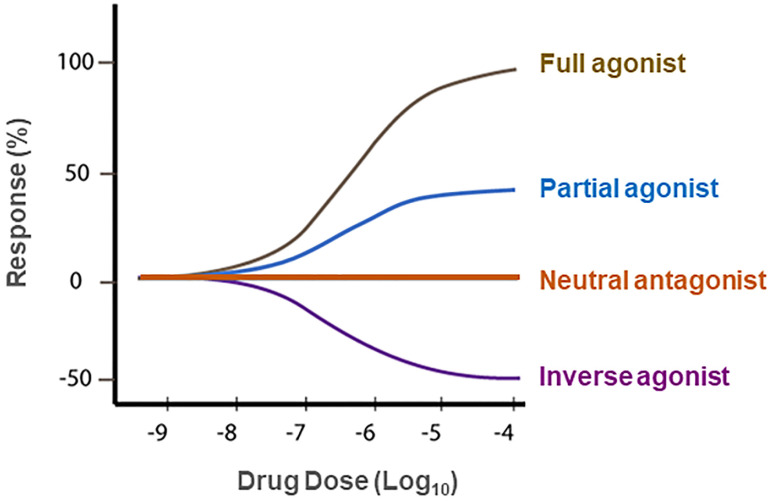
Concepts of neutral antagonist vs. inverse agonist. A **full agonist** (such as WIN55,212-2 or CP55,940) binds to and activates a receptor, producing maximal (100%) biological effects. A **partial agonist** (such as Δ^9^-THC) produces an effect that is less than a full agonist. An **inverse agonist** (such as rimonabant in the absence of CB1 receptor agonist) binds to the same receptor as an agonist but produces a biological effect opposite to that of an agonist. A **neutral antagonist** (such as AM4113 or PIMSR) binds to a receptor and blocks agonist binding to the same receptor, while itself does not produce any effect in the absence of an agonist. At basal conditions, a receptor has intrinsic activity or is under a balance between active and inactive states in the absence of endogenous ligands. An agonist increases the activity of a receptor above its basal level, whereas an inverse agonist decreases the activity below the basal level.

**Figure 4 cells-11-03262-f004:**
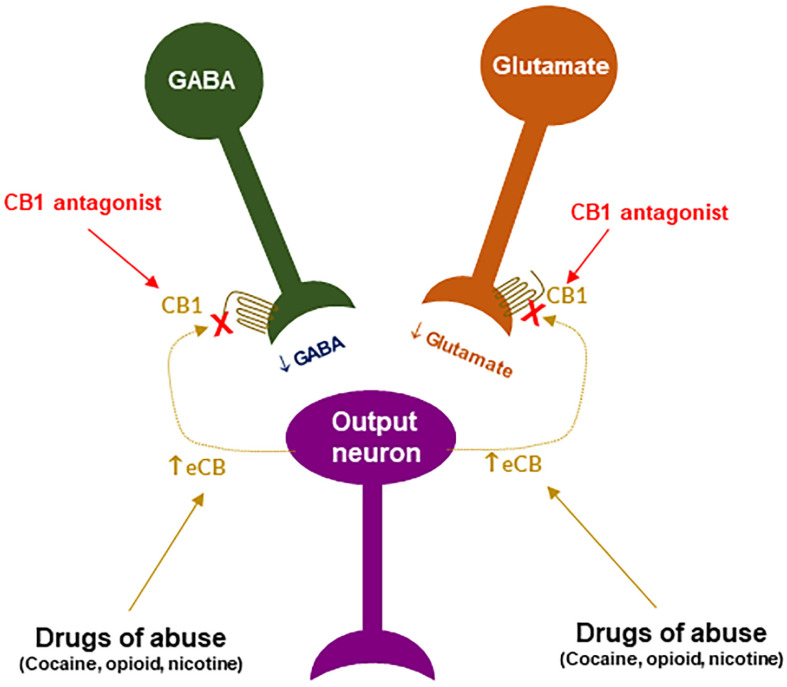
A schematic diagram of an endocannabinoid (eCB) hypothesis showing how a CB1R antagonist blocks actions produced by drugs of abuse. Growing evidence indicates that drugs of abuse (such as cocaine, opioid or nicotine) may stimulate endocannabinoid (such as 2-AG, AEA) release from postsynaptic neurons (such as VTA DA neurons), which subsequently activates CB1R on presynaptic CB1Rs and produces a reduction in GABA or glutamate release. A CB1R antagonist, including neutral antagonist, binds to CB1R and blocks eCB binding to the same receptor, therefore producing antagonism of drug reward and relapse. ↑: indicates an increase in eCB release; “×” indicates blockade of CB1 receptor.

**Figure 5 cells-11-03262-f005:**
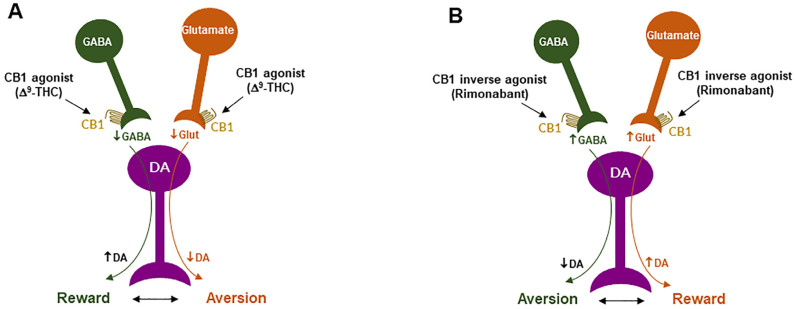
Schematic diagrams of a working hypothesis showing how a CB1R agonist and an inverse agonist produce opposite subjective effects. (**A**): A CB1R agonist such as Δ^9^-THC may produce rewarding effects by decreasing VTA GABA release and increasing NAc DA release. (**B**): In contrast, an inverse CB1R agonist may produce an opposite aversive effect by increasing GABA release and decreasing DA release in the NAc. ↑, ↓: indicate an increase or a decrease in neurotransmitter release.

**Table 1 cells-11-03262-t001:** The literature reports regarding the rewarding vs. aversive effects of SR141716 itself in rodents.

Compound	Doses	Species	Results	References
Rimonabant	20 mg/kg, i.p.	Rat	↓ Electrical brain-stimulation reward	[142]
Rimonabant	0.3, 1, 3, 10 mg/kg, i.p.	Rat	↓ Electrical brain-stimulation reward	[139]
Rimonabant	0.02, 0.3, 1.0 mg/kg, i.p.	Rat	No effect on electrical brain-stimulation reward	[144]
Rimonabant	0.02 mg/kg, i.p.	Rat	No effect on electrical brain-stimulation reward	[145]
Rimonabant	3, 10 mg/kg, i.p.,	Mouse	No effect on electrical brain-stimulation reward	[143]
Rimonabant	0.3, 1, 3 mg/kg, i.p.	Rat	Not produce CPP or CPA	[127]
Rimonabant	0.1, 0.5, 3.0 mg/kg, i.p.	Rat	Not produces CPP or CPA	[148]
Rimonabant	0.5, 1, 2 mg/kg, i.p.	Rat	Not produce CPP or CPA	[149]
Rimonabant	3 mg/kg, i.p.	Rat	Not produce CPP or CPA	[89]
Rimonabant	0.25, 0.5, 1 mg/kg, i.p.	Rat	Not produce CPP or CPA	[55]
Rimonabant	0.1, 0.5, 3 mg/kg	Rat	Not produce CPP or CPA	[150]
Rimonabant	0.3, 3 mg/kg, i.p.	Rat	Not produce CPP or CPA	[151]
Rimonabant	0.25, 0.5, 2, 3 mg/kg	Rat	Produces CPP	[53]
Rimonabant	3 mg/kg	Mouse	↓ Accumbens DA	[146]
Rimonabant	2, 10 mg/kg, i.p.	Rat	No effect on accumbens DA	[76]
Rimonabant	1, 10, 30, 100 mM, intra-NAc	Rat	↑ Accumbens DA	[76]
Rimonabant	5, 10, 20 mg/kg, i.p.	Rat	↑ Accumbens DA	[147]

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
