# Peer review of "Neutral CB1 Receptor Antagonists as Pharmacotherapies for Substance Use Disorders: Rationale, Evidence, and Challenge"

_cells, 2022, doi:10.3390/cells11203262_

Round 1

Reviewer 1 Report

Soler-Cedeno and Xi provide a comprehensive overview of the progression of the development of neutral CB1 receptor agonists. The authors introduce the topic and provide a rationale for the use of neutral CB1, with a clear connection to the physiology of SUDs, the endocannabinoid system and the brain. Current knowledge regarding cannabinoids and cannabis reward/aversion are highlighted. The authors also provide a useful overview of the history of rimonabant and associated mechanisms of action, while clearly highlighting the issues that led to the withdrawal of the drug from the market. The evidence supporting CB1R antagonism for the treatment of SUDs is summarized, including a clear explanation of how the inverse agonist activity of many substances may hinder their use clinically. Finally, the authors discuss the future use of neutral antagonists for the treatment of SUDs, with several specific examples.

Though a fair amount of literature exists regarding the legacy of rimonabant and neutral CB1 antagonists, most of the summative papers are relatively dated. As this review provides an update on the current state of the field and incorporates recent findings, it represents a valuable addition to the literature. Some minor changes should be made to improve the quality of the document of publication, as indicated in the following section. 

Specific Notes:

·       There is a significant amount of self-citation in the introduction of the paper. It would be better to provide the original scientific source. For example, the original data papers regarding rimonabant should be cited.

·       It would be useful to provide a definition of psychostimulant use disorder.

·       In section 2, you may find it helpful to clearly indicate a comprehensive review that the reader may refer to for more information. The figure provides a good overview of the topics discussed.

·       Some references regarding the 2-AG in the brain would lend support to the idea that it is a major player in modulating brain function.

·       A reference should be provided regarding the binding affinities of the phytocannabinoids and synthetic cannabinoids.

·       A reference is needed regarding the prevalence of cannabis use globally.

·       The discussion on the paradoxical effects of cannabinoids is somewhat unclear. Are the effects related to differences in species? There is some discussion later of how cannabinoids are aversive rather than rewarding later in the paper. Relating this back to amotivational syndrome might be beneficial.

·       In section 5.1, additional supporting references should be provided that are not self-citation. If no other evidence exists, this should be clearly indicated.

·       When describing opposing results in the studies regarding CB1 activation and DA release/neuronal firing, it would be helpful to indicate species.

·       Some of the detail provided in section 5.2 is not needed based on the focus of the paper.

·       Consistency with drug naming (ex. Rimonabant vs. SR141716A) would improve clarity

·       In the section discussing the pharmacological effects of rimonabant, it is unclear what is unexpected about a CB1R antagonist producing opposite effects of an agonist. It becomes more clear later in the paragraph that you are attempting to situate rimonabant as both an antagonist and an inverse agonist, but this paragraph should be edited for clarity. This can be accomplished by discussing the dual action of rimonabant and then providing the evidence to support this.

·       The definitions and figures provided in the paper were very helpful for breaking down pharmacologic concepts.

·       Some extraneous detail is provided in the section detailing the specific neutral antagonists. It would be sufficient to provide brief summaries of the important findings associated with each compound and direct the reader to the paper of interest rather than relaying all the findings from previous papers published. This is an issue, particularly with PIMSR.

·       The paper would benefit from some editing for grammar and spelling.

·       Many of the paragraphs could benefit from the addition of a concluding sentence to wrap up what has been discussed in the paragraph and bridge to the next section. 

Author Response

Please see the attached file "Author response to reviewers" below

Reviewer 2 Report

This work by Soler-Cerdeno & Xi is a critical, extensive and well-written review about the role of neutral CB1r antagonists as pharmacotherapies for substance use disorders. In my opinion, it is a very helpful and detailed review, although there are some concerns to be addressed by the authors:

- In the section “2. Mesocorticolimbic dopamine system”, it should be added some brief mention to the 3 stages cycle of addiction, as well as include alcohol since it is the most consumed drug worldwide (this also applies to Figure 4).

- In section “4. Endocannabinoid system”, please reorganize to first comment all the main components of the system, and then include the information in this order: 1) cannabinoid receptors (which were the first to be identified in the 80s), 2) endocannabinoid ligands, 3) synthesis and degradation enzymes. It is important to extend here the information about CB1r, since is the main focus of the review (localization, expression level, functions, etc.).

- I am not sure about the rationale to include section “5. GABAergic CB1R hypothesis of cannabis reward”. Please consider to significantly reduce the contents, or integrate them in the previous section about the endocannabinoid system. In addition, be careful when stating “CB2Rs are expressed predominantly in peripheral tissues”. In spite this is not the focus of the revision, nowadays there is extensive literature about the presence and the functional role of CB2r in neurons, astrocytes and microglia.

- Several clinical trials aimed to evaluate the anti-obesity properties of rimonabant suggested an association with the appearance of depressive symptoms, suicidal ideation or, at worst, suicide (Christensen, Kristensen, Bartels, Bliddal & Astrup, 2007). However, it is essential to point out that some confounders could be implicated, such as the inclusion of obese patients with a previous history of depression, a fact that could increase the possibility of detecting depressive symptomatology, including suicidal ideation or suicide in some patients treated with rimonabant (Bluher, 2008; Despres, Van Gaal, Pi-Sunyer & Scheen, 2008). Thus, this is crucial to be mentioned by authors in the manuscript.

 References:

Christensen R, Kristensen PK, Bartels EM, Bliddal H, & Astrup A (2007). Efficacy and safety of the weight-loss drug rimonabant: a meta-analysis of randomised trials. Lancet 370: 1706-1713.

Bluher M (2008). Efficacy and safety of the weight-loss drug rimonabant. Lancet 371: 555-556; author reply 556-557.

Despres JP, Van Gaal L, Pi-Sunyer X, & Scheen A (2008). Efficacy and safety of the weight-loss drug rimonabant. Lancet 371: 555; author reply 556-557.

Author Response

Please see the attached file named "Author response to reviewers" below.
